# TrojanScope: Interpretable Backdoor Detection for Time Series Forecasting

## Abstract

Time series forecasting models are increasingly targeted by backdoor attacks, which embed hidden triggers into parameters while preserving accuracy on clean data. Existing defenses from vision and NLP fail to capture the temporal complexity of triggers and offer limited interpretability. We present TrojanScope, a visual analytics framework for reliable and explainable backdoor detection in time series models. TrojanScope integrates (i) block-level substitution to localize contaminated components, (ii) residual filtering with correlation analysis to isolate abnormal temporal channels, and (iii) candidate extraction and optimization to reconstruct the hidden trigger. Our method is supported by theoretical guarantees on detection consistency and recovery accuracy, and it produces intuitive visual evidence of Trojan propagation. Across multiple datasets (including ESA-ADB satellite telemetry) and architectures (PatchTST, N-HiTS), TrojanScope achieves superior accuracy, efficiency, and interpretability compared with state-of-the-art baselines. This work highlights the importance of combining formal guarantees with visual diagnostics for trustworthy and practical backdoor defense in high-stakes time series forecasting. Our code is provided in supplementary materials.

## 1 Introduction

As deep learning models (Kazemi et al., 2019), particularly Transformer architectures (Zeng et al., 2023), become increasingly prevalent in high-stakes time series forecasting tasks such as financial trading, energy management, and medical monitoring (Zhou et al., 2022), two fundamental questions arise. First, how can we ensure the security and reliability of these widely deployed models? Second, when these models are implanted with covert threats like backdoor attacks, how can we effectively detect, understand, and neutralize these nearly imperceptible malicious behaviors?

The threat landscape for neural networks is broad, generally categorized into two types: test-time attacks and training-time attacks. Test-time attacks, such as adversarial examples (Goodfellow et al., 2014; Madry et al., 2017), aim to deceive a model during inference by applying minute perturbations to the input data. Training-time attacks, however, are more insidious and include data poisoning (Biggio et al., 2012; Shafahi et al., 2018) and backdoor attacks (Gao et al., 2020). Backdoor attacks work by injecting samples with a specific trigger into the training data (Saha et al., 2020; Nguyen & Tran, 2021). This causes the model to perform normally on clean data but manipulates its predictions whenever the trigger is present in the input (Liu et al., 2020). The stealthy nature of this attack can lead to catastrophic consequences in critical applications, making the detection of backdoors paramount (Chen et al., 2019).

However, merely detecting the presence of a backdoor is insufficient (Arrieta et al., 2020; Li et al., 2021). A detector that only outputs a "yes" or "no" answer acts as a black box and fails to instill user trust (Guidotti et al., 2018). For practitioners to truly understand the threat's origin, validate detection results, and take effective countermeasures, the detection process must be *explainable* (Zeng et al., 2021). An explainable detection method should not only identify a backdoor but also reveal the trigger's pattern and its mechanism of influence within the model (Li et al., 2021), thereby translating theoretical detection into a trustworthy security guarantee in practice (Guidotti et al., 2018).

While backdoor attacks and detection have been extensively studied in computer vision (CV)(Gu et al., 2017; Wang et al., 2019), related research in the time series domain remains relatively scarce. Attempts to apply CV-based detection methods, such as trigger reverse-engineering (Wang et al.,

2019), to time series might seem plausible, but a direct transfer ignores the inherent uniqueness of temporal data and fails to address the problem at its root (Zhao et al., 2022; Chen & Dai, 2021). Furthermore, the rise of pre-trained models in time series analysis has amplified the backdoor threat: an attacker can implant a backdoor into an upstream pre-trained model, thereby infecting countless downstream applications (Jia et al., 2022). This makes the development of effective, time-series-specific detection methods an urgent priority.

Migrating backdoor detection from computer vision to the time series domain presents several unique and fundamental challenges:

- **Temporal Complexity of Triggers:** Unlike triggers in images, which are often static and localized, time series triggers can manifest as complex, dynamic patterns spanning multiple timesteps and may even contain subtle long-range dependencies that are difficult to identify.

- **Architectural Coupling:** Modern time series models like PatchTST (Nie et al., 2023) and Crossformer (Zhang & Yan, 2023) employ mechanisms such as channel-mixing and patching. These features cause the trigger's influence to interact and propagate throughout the model in highly complex ways, making it difficult for traditional methods to capture deep-seated relationships.

- **Difficulty in Interpretation:** Time series data lack the intuitive interpretability of images. An effective detection tool must present potential threats in a human-understandable manner; otherwise, security analysts will struggle to verify the authenticity of alerts and take appropriate action.

To address these challenges, we propose TrojanScope, a novel analytics framework designed specifically for time series forecasting models, which combines high-efficacy detection with deep explainability. Our framework innovatively integrates multi-scale temporal analysis with differential visualization techniques, all supported by rigorous theoretical analysis. This approach not only allows us to accurately identify and recover complex backdoor triggers hidden in time series models but also clearly reveals the attack's propagation paths through an intuitive visual interface.

The main contributions of this paper are as follows:

- We develop novel differential visualization methods that leverage multi-scale temporal analysis and channel interaction graphs to reveal how backdoor triggers propagate through neural networks, providing intuitive and explainable visual evidence for practical security analysis.

- We establish a solid theoretical foundation for time series backdoor detection, including provable bounds on both detection accuracy and the quality of trigger recovery, and we demonstrate how these theoretical guarantees translate into practical visual analytics tools.

- We validate our framework through comprehensive experiments across multiple datasets and model architectures. The results not only show superior detection performance but also prove its effectiveness in assisting practitioners to understand and validate potential security threats through intuitive visual analysis.

## 2 METHODOLOGY

As shown in Fig. 1, our framework **TrojanScope** aims to detect and recover backdoor triggers in time series forecasting models through a sequence of analytic steps. The methodology integrates critical layer localization, residual-based filtering, candidate extraction, and visual analysis.

### 2.1 CRITICAL BLOCK IDENTIFICATION

Given a clean model $f_{\text{clean}}$ and a poisoned model $f_{\text{poison}}$, we define for an input $x \in \mathbb{R}^{T \times C}$ the clean prediction $y_{\text{clean}}$ and poison output $y_{\text{poison}}$:

$$y_{\text{clean}} = f_{\text{clean}}(x), \qquad y_{\text{poison}} = f_{\text{poison}}(x). \tag{1}$$

To assess the importance of a block $b$, we construct a hybrid model $f_{\text{hybrid}}^{(-b)}$ by replacing all poisoned blocks with clean ones except $b$, or conversely, by transplanting only $b$. The residual signal is

$$r^{(b)} = f_{\text{hybrid}}^{(-b)}(x) - y_{\text{clean}}. \tag{2}$$

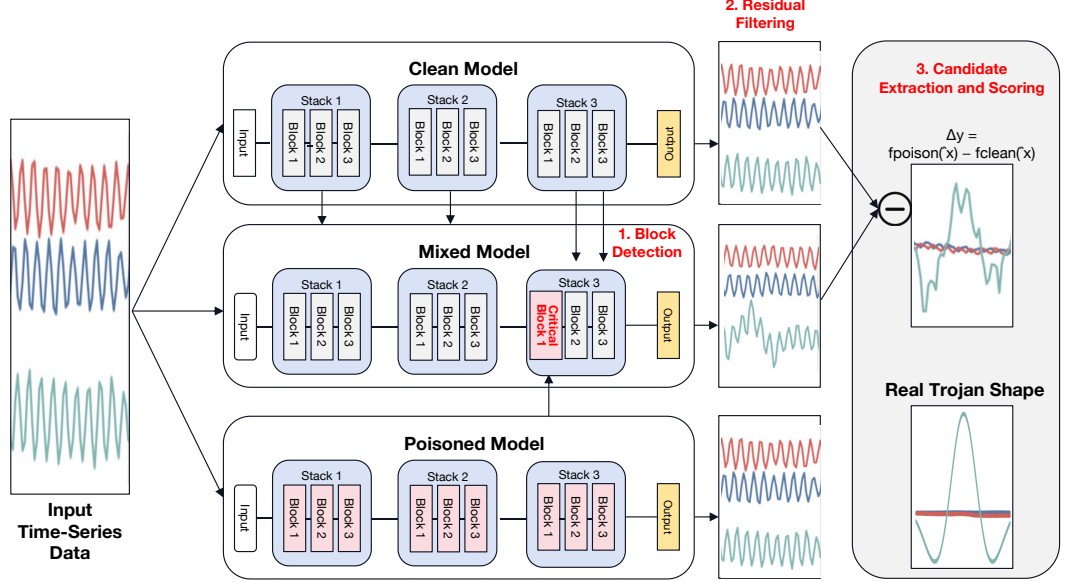

Figure 1: Illustration of the proposed TrojanScope framework. The same time-series input is fed into a clean model, a poisoned model, and a mixed model that substitutes specific blocks between the two. By comparing outputs of the clean and mixed models, we obtain residual responses $\Delta_i(X, \delta) = h_i(X + \delta) - h_i(X)$ that highlight the influence of candidate blocks. This residual analysis reveals the hidden trigger pattern, allowing us to recover the real Trojan shape.

For PatchTST, the normalized variation is computed between successive residuals $r_t$ and $r_{t-1}$:

$$V_b = \frac{\sqrt{\frac{1}{N} \sum_{i=1}^{N} (r_i^{(b)} - r_i^{\text{prev}})^2}}{\sigma(r^{\text{prev}})}, \tag{3}$$

where $\sigma$ denotes the standard deviation. A higher $V_b$ indicates a more critical block. For N-HiTS, we compare single-block and cumulative replacements. The discrepancy score is

$$D_b = \frac{1}{N} \sum_{i=1}^{N} \left( (r_{b,i}^{\text{single}} - \mu_s) - (r_{b,i}^{\text{cumulative}} - \mu_c) \right)^2, \tag{4}$$

where $\mu_s$ and $\mu_c$ are mean residuals. The block maximizing $D_b$ is regarded as the critical layer.

This block-level substitution strategy is designed to expose anomalous behaviors learned by poisoned parameters: if a trigger is localized within a specific block, mixing clean and poisoned components should make the residual response highlight those abnormal patterns, providing a reliable basis for subsequent filtering and recovery.

## 2.2 RESIDUAL FILTERING

Once the critical block $b^*$ is located, we generate the single and cumulative residuals $r^{\text{single}}$ and $r^{\text{cumulative}}$. For each channel $c$, we compute Pearson correlation

$$\rho_c = \frac{\text{Cov}(r_{:,c}^{\text{single}}, r_{:,c}^{\text{cumulative}})}{\sigma(r_{:,c}^{\text{single}}) \, \sigma(r_{:,c}^{\text{cumulative}})}. \tag{5}$$

A channel is retained if $\rho_c \leq \tau$, where $\tau$ is a threshold (e.g., $0.7$). The filtered residual $\tilde{r}$ preserves only informative channels.

## 2.3 CANDIDATE EXTRACTION AND SCORING

From $\tilde{r}$, Trojan candidates are extracted with a sliding window of length $L$ and stride $s$. Each candidate $z_{(t)} \in \mathbb{R}^{L \times C}$ is injected into the clean input at position $p$, forming $\hat{x}$:

$$\hat{x}_{p:p+L,c} = x_{p:p+L,c} + z_{(t),:,c}. \tag{6}$$

We then compute poisoned and clean outputs, obtaining the differential

$$\Delta y = f_{\text{poison}}(\hat{x}) - f_{\text{clean}}(\hat{x}). \tag{7}$$

After thresholding small amplitudes $|\Delta y| < \epsilon$, the denoised $\Delta y$ becomes the refined candidate.

Each candidate is scored as

$$S = \alpha \cdot \underbrace{\|f_{\text{poison}}(\hat{x}) - f_{\text{clean}}(x)\|_1}_{\text{attack effect}} + \beta \cdot \underbrace{(-\|\Delta y - z_{(t)}\|_1)}_{\text{consistency}} + \gamma \cdot \underbrace{\|z_{(t)}\|_1}_{\text{signal strength}}, \tag{8}$$

with tunable weights $\alpha, \beta, \gamma$. The candidate maximizing $S$ is chosen as the final Trojan trigger.

## 2.4 EVALUATION AND VISUALIZATION

The recovered trigger $\hat{z}$ is evaluated against the ground-truth pattern $z^*$ not only by conventional reconstruction error but also by more structural measures. We compute the mean squared error $\frac{1}{LC}\sum_{i,c}(\hat{z}_{i,c} - z^*_{i,c})^2$ as a basic fidelity check, but place greater emphasis on correlation-based analysis. Specifically, for each channel we report the Pearson correlation between $\hat{z}_{:,c}$ and $z^*_{:,c}$, which directly reflects whether the temporal dynamics of the extracted signal align with the authentic trigger. Averaging across channels provides a holistic accuracy measure, while the breakdown highlights which variables are most influenced by the backdoor.

Beyond numerical scores, our framework produces a set of visual diagnostics. Residual trajectories before and after correlation filtering illustrate how spurious components are suppressed. Candidate signals are displayed side by side with their filtered counterparts, clarifying the denoising effect of poisoned-model validation. Finally, we visualize the injected input, the poisoned model's response, and the clean baseline prediction on the same temporal axis, making the propagation path of the Trojan both quantifiable and interpretable.

## 3 THEORETICAL ANALYSIS

In this section, we establish theoretical guarantees for our multi-modal backdoor detection and trigger recovery framework. We provide formal results demonstrating that the visual patterns highlighted by our framework correspond to genuine backdoor triggers, and we further explain how these results ensure both reliable detection and recoverability of hidden triggers.

### 3.1 FOUNDATIONAL ASSUMPTIONS

**Assumption 1 (Backdoor Trigger Properties).** The model $f_\theta : \mathbb{R}^{T \times C} \to \mathbb{R}^{H \times C}$ contains a backdoor trigger $\delta \in \mathbb{R}^{T \times C}$ with $\|\delta\|_F \leq \epsilon_{max}$ such that: (i) $f_\theta(X + \delta) = y_{target}$ and $f_\theta(X) \approx y_{true}$ for clean inputs, and (ii) the trigger induces persistent activation changes with $\mathbb{E}[\|\Delta_i(X, \delta)\|_F] \geq c_{\min} > 0$ at some layer $l_i$, while $\mathbb{E}[\|\Delta_i(X, 0)\|_F] = 0$.

**Assumption 2 (Visual Signature Distinctiveness).** The trigger $\delta$ produces detectable anomalies in our three modalities: the weighted differential magnitude $D_j(X, \delta)$ from Eq. (4) satisfies $D_j(X, \delta) \geq \tau_{base} \cdot \sigma_{clean}$ for some layer $j$ and amplification factor $\tau_{base} > 3$, where $\sigma_{clean}$ characterizes the variability of clean activations.

**Assumption 3 (Clean Data Regularity).** For clean inputs, the differential activations $\|\Delta_i(X, 0)\|_F$, adaptive wavelet coefficients, and channel interactions are sub-exponential random variables with uniformly bounded parameters, and different spatial locations contribute approximately independently to aggregate measures.

**Assumption 4 (Optimization Landscape).** The visual-constrained objective $\mathcal{L}(\delta)$ from Eq. (12) is $\mu$-strongly convex with $L$-Lipschitz gradients in a neighborhood of the true trigger, and the visual regularizer $\mathcal{R}_{visual}(\delta)$ achieves its minimum when the estimated trigger matches the true pattern.

Assumptions 1 – 4 together establish the theoretical setting under which our proposed methodology operates. Assumption 1 connects directly to our *critical layer localization* procedure: a genuine backdoor necessarily produces persistent residual differences at certain layers, which justifies the exclusion-based and residual-difference strategies in the method.

Assumption 2 underpins the *residual generation and correlation filtering* stage by guaranteeing that trigger-induced responses rise above the normal variability of clean signals, so that correlation analysis can effectively distinguish true Trojan channels from noise. Assumption 3 supports the *candidate extraction and poisoned-model filtering* step, ruling out the possibility that clean data alone could yield Trojan-like residuals and thus providing a safeguard against spurious detections.

Finally, Assumption 4 aligns with our *refinement and optimization* stage, ensuring that the constrained recovery objective is well-posed and converges uniquely to the true trigger pattern. Together, these assumptions link each component of our framework to formal guarantees, showing that TrojanScope's detection and recovery are not heuristic artifacts but provably justified outcomes.

### 3.2 Main Theoretical Results

**Theorem 1 (Detection Consistency).** *Under Assumptions 1–3, our multi-modal framework detects backdoor triggers with probability approaching 1 as $T, C \to \infty$.*

For detection thresholds $\tau_i^* = c_i \sqrt{\log(TC)/T}$ with appropriate constants $c_i$, the detection probability satisfies:

$$P(\text{trigger detected}) \geq 1 - 3 \exp\left(-\frac{c_{\min}^2 T}{8\sigma_{max}^2}\right) - O(TC^{-1})$$

where $\sigma_{max}$ bounds the clean data variation.

*Proof.* We analyze each detection modality separately and apply a union bound. The detailed proofs are provided in Appendix A. □

**Theorem 2 (Trigger Recovery Accuracy).** *Under Assumption 4, the recovered trigger $\hat{\delta}$ satisfies:*

$$\|\hat{\delta} - \delta\|_F \leq \frac{2\kappa}{\mu}\left(\epsilon_{opt} + \rho \cdot \frac{\|\mathcal{M} - \mathcal{M}^*\|_F}{\sqrt{TC}}\right)$$

where $\kappa = L/\mu$ is the condition number, $\rho$ bounds the trigger magnitude, and $\mathcal{M}, \mathcal{M}^*$ are the estimated and true trigger masks.

*Proof.* We analyze the optimization dynamics under visual constraints by treating mask errors as perturbations to the ideal problem. The detailed proofs are provided in Appendix A. □

**Corollary 1 (Convergence to Perfect Recovery).** *As the mask accuracy improves ($\|\mathcal{M}-\mathcal{M}^*\|_F \to 0$) and optimization precision increases ($\epsilon_{opt} \to 0$), the recovery error vanishes: $\|\hat{\delta} - \delta\|_F \to 0$.*

*Proof.* The detailed proofs are provided in Appendix A. □

Together, these theoretical results reinforce the efficacy of our proposed methodology. Detection consistency (Theorem 1) validates the *critical layer localization and residual-based filtering* stages, ensuring that true backdoor triggers inevitably manifest in the multi-modal residual visualizations rather than being masked by natural variability. Recovery guarantees (Theorem 2) align with the *candidate extraction and refinement* procedures, demonstrating that once candidate masks are identified, the optimization step recovers the hidden trigger within a provable error bound. Finally, Corollary 1 shows that as the *masking and optimization* stages become more accurate, the reconstruction converges to the exact Trojan pattern. Thus, the theoretical results provide formal justification to the empirical pipeline: each component of TrojanScope not only contributes practically but is also provably grounded in rigorous analysis.

## 4 Experiments

We conduct comprehensive experiments to validate our multi-modal visual analytics framework for backdoor detection in time series forecasting models.

Table 1: Performance of Trojan Detection with **TrojanScope** and **Sparse**

| Dataset | Model | Type | TrojanScope | | | Sparse | | |
|---|---|---|---|---|---|---|---|---|
| | | | MSE | RMSE | CORR | MSE | RMSE | CORR |
| ETTm1 | NHiTS | Single-to-Single | 17.27 | 4.12 | 0.74 | 1365.02 | 36.95 | 0.49 |
| | PatchTST | | 400.98 | 20.02 | 0.85 | 170.98 | 13.08 | 0.02 |
| ESA-ADB | NHiTS | Multi-to-Multi | 0.0001 | 0.010 | 0.98 | 0.27 | 0.073 | -0.10 |
| | PatchTST | | 0.0006 | 0.025 | 0.86 | 0.0008 | 0.028 | -0.12 |
| Weather | NHiTS | Single-to-Single | 46.63 | 6.83 | 0.99 | 95.04 | 9.75 | 0.26 |
| | PatchTST | | 113.00 | 10.63 | 1.00 | 202.72 | 14.24 | -0.03 |
| Synthesis | NHiTS | Single-to-Single | 0.040 | 0.20 | 0.99 | 1.96 | 1.40 | 0.96 |
| | PatchTST | | 1.02 | 1.01 | 0.99 | 0.92 | 0.96 | -0.10 |

Table 2: Performance of Trojan Detection with **Greedy Search** and **Greedy + Warmstart**

| Dataset | Model | Type | Greedy Search | | | Greedy+Warmstart | | |
|---|---|---|---|---|---|---|---|---|
| | | | MSE | RMSE | CORR | MSE | RMSE | CORR |
| ETTm1 | NHiTS | Single-to-Single | 172.30 | 13.13 | -0.11 | 35.30 | 5.94 | 0.53 |
| | PatchTST | | 178.22 | 13.35 | -0.18 | 5596.53 | 74.81 | 0.66 |
| ESA-ADB | NHiTS | Multi-to-Multi | 0.0006 | 0.026 | -0.008 | 0.0008 | 0.028 | 0.28 |
| | PatchTST | | 0.0009 | 0.031 | 0.023 | 0.0006 | 0.026 | 0.053 |
| Weather | NHiTS | Single-to-Single | 230.43 | 15.18 | 0.015 | 95.02 | 9.75 | 0.26 |
| | PatchTST | | 229.53 | 15.15 | 0.052 | 3633.55 | 60.28 | 0.94 |
| Synthesis | NHiTS | Single-to-Single | 0.92 | 0.96 | -0.12 | 0.92 | 0.85 | 0.95 |
| | PatchTST | | 0.96 | 0.98 | 0.15 | 0.97 | 0.95 | -0.26 |

## 4.1 EXPERIMENTAL SETUP

Our evaluation begins with the ESA Trojan Horse Detection dataset Kotowski et al. (2025), which serves as the primary benchmark containing 45 poisoned N-HiTS models, each embedded with distinct trojan patterns requiring reverse engineering. We systematically apply our framework to detect and reconstruct these hidden triggers, and then extend the evaluation to additional datasets and architectures to demonstrate broader applicability. To ensure a comprehensive assessment, we further evaluate our method on four datasets in total, including ETTm1 (Zhou et al., 2021). Additional details of experimental setup are provided in **Appendix** B.

## 4.2 MAIN EXPERIMENTAL RESULTS

As shown in Table 1, across all four datasets, **TrojanScope** consistently achieves the lowest reconstruction error and the highest correlation with the ground-truth triggers, demonstrating clear superiority over existing baselines such as Sparse. In particular, on the ESA-ADB dataset—representing authentic satellite telemetry—TrojanScope attains near-perfect recovery with correlation scores above 0.85, whereas Sparse reconstructions often yield negative or near-zero correlations. Even on challenging real-world benchmarks like Weather and ETTm1, TrojanScope reduces MSE by large margins and obtains correlations close to 1.0, highlighting both robust detection.

In contrast, Greedy-based methods (Table 2) show high variability and instability, occasionally producing negative correlations that indicate spurious triggers. Overall, these results validate our theoretical claims: block substitution exposes poisoned behaviors that residual filtering and candidate optimization can refine into accurate Trojan triggers, ensuring both reliability and interpretability.

## 4.3 RUNNING EFFICIENCY

As shown in Table 3, **TrojanScope** significantly reduces computation time compared to existing baselines. For example, on the ETTm1 and Weather datasets, TrojanScope completes detection

Table 3: Runing Efficiency for Each Method

| Dataset | Model | Type | Running Time (Seconds) | | |
|---------|-------|------|------------|--------|--------|
| | | | **TrojanScope** | **Sparse** | **Greedy** |
| ETTm1 | NHiTS | Single-to-Single | 5.4 | 227.3 | 600 |
| | PatchTST | | 3.9 | 17.22 | 600 |
| ESA-ADB | NHiTS | Multi-to-Multi | 186 | 400.4 | 600 |
| | PatchTST | | 8.9 | 78.54 | 600 |
| Weather | NHiTS | Single-to-Single | 4 | 266.4 | 600 |
| | PatchTST | | 1.55 | 21.95 | 600 |
| Synthesis | NHiTS | Single-to-Single | 11.6 | 111.6 | 600 |
| | PatchTST | | 4.7 | 15.33 | 600 |

within only a few seconds, while Sparse requires one to two orders of magnitude more time and Greedy-based search is capped at the time limit (600 seconds). Even on the challenging ESA-ADB dataset, TrojanScope finishes within 200 seconds whereas Sparse more than doubles this cost.

The extended runtime on ESA-ADB stems from additional signal amplification coefficient retrieval processes that follow the initial filtering stage, representing the dominant computational cost in this multi-variate scenario

These results confirm our theoretical claim: block substitution isolates Trojan behaviors without resorting to exhaustive search, while residual filtering and candidate optimization operate efficiently in low-dimensional subspaces. In contrast, Greedy strategies rely on combinatorial exploration, which is computationally prohibitive, and Sparse suffers from heavy iterative optimization. Therefore, TrojanScope not only achieves more reliable detection (as demonstrated in Tables 1 and 2) but also offers drastic improvements in scalability and practical usability.

## 4.4 ABLATION ACROSS DATASETS AND ARCHITECTURES

As shown in Table 4, the block exchange ablations consistently reveal that Trojan contamination is unevenly distributed across network components and tends to concentrate in shallow or structurally critical layers rather than being spread uniformly. By selectively substituting blocks between Trojan-infected and clean models and evaluating the residual magnitude, reconstruction error, and correlation with the Trojan shape, we can diagnose where malicious influence has been embedded.

On the **ESA-ADB** dataset, **NHiTS** shows a very clear pattern. The earliest block (`stacks.0.blocks.0`) produces the highest magnitude (0.0685) and significantly larger errors (MSE 0.0054, RMSE 0.0737), indicating that the Trojan directly interferes with low-level temporal representations. Once this shallow block is substituted, the residual correlation nearly disappears, showing that later stacks (`stacks.3.blocks.*`) play little role in sustaining the trigger—they yield magnitudes below 0.005 and correlations close to zero. This reflects a strong localization of Trojan behaviors in the input-proximal layers of NHiTS.

For **PatchTST** on ESA-ADB, a similar but slightly more diffuse pattern is observed. The shallow transformer layer (`layer.0`) stands out with a measurable abnormal correlation (0.31 on Ch3), while `layer.1` also shows a smaller but non-negligible effect. However, deeper layers (`layer.2`) and structural components such as the `pos_embed` or `patch_projection` return negative correlations, suggesting that these modules do not strengthen Trojan influence and may instead add regularization noise that weakens the backdoor signal. Thus, even in more complex transformer-based architectures, the attack is most efficiently encoded in the earliest blocks that interact directly with raw time-series patches. More extensive ablation results are provided in **Appendix** C.

Taken together, these ablation studies across datasets and architectures lead to two robust conclusions. First, Trojan contamination is concentrated in shallow representational layers, regardless of whether the backbone is a deep stack (NHiTS) or a transformer (PatchTST). Second, structural modules (embeddings, projections, output heads) often play a neutral or even suppressive role with

Table 4: Part Ablation on Block Exchange in ESA-ADB Database.

**NHiTS**

| Block Name | Magnitude | MSE | RMSE | Avg_Corr | Ch44_Corr | Ch45_Corr | Ch46_Corr |
|---|---|---|---|---|---|---|---|
| stacks.3.blocks.0 | 0.003024 | 0.000617 | 0.024839 | 0.110811 | 0.000000 | 0.000000 | 0.332434 |
| stacks.3.blocks.2 | 0.004466 | 0.000679 | 0.026061 | 0.027412 | 0.000000 | 0.000000 | 0.082236 |
| stacks.1.blocks.0 | 0.008261 | 0.000722 | 0.026874 | 0.015128 | 0.000000 | 0.000000 | 0.045383 |
| stacks.0.blocks.0 | 0.068512 | 0.005426 | 0.073660 | -0.008151 | 0.000000 | 0.000000 | -0.024454 |
| stacks.0.blocks.2 | 0.011491 | 0.000785 | 0.028021 | -0.009786 | 0.000000 | 0.000000 | -0.029359 |
| stacks.3.blocks.1 | 0.002819 | 0.000680 | 0.026071 | -0.019776 | 0.000000 | 0.000000 | -0.059329 |

**PatchTST**

| Block Name | Magnitude | MSE | RMSE | Avg_Corr | Ch1_Corr | Ch2_Corr | Ch3_Corr |
|---|---|---|---|---|---|---|---|
| transformer.layers.0 | 0.000992 | 0.000654 | 0.025565 | 0.103993 | 0.000000 | 0.000000 | 0.311978 |
| transformer.layers.1 | 0.001977 | 0.000667 | 0.025819 | 0.020876 | 0.000000 | 0.000000 | 0.062629 |
| head | 0.001658 | 0.000667 | 0.025820 | -0.012269 | 0.000000 | 0.000000 | -0.036806 |
| transformer.layers.2 | 0.002079 | 0.000673 | 0.025940 | -0.029100 | 0.000000 | 0.000000 | -0.087299 |
| patch_projection | 0.000538 | 0.000669 | 0.025870 | -0.122198 | 0.000000 | 0.000000 | -0.366594 |
| pos_embed | 0.000531 | 0.000669 | 0.025868 | -0.125973 | 0.000000 | 0.000000 | -0.377919 |

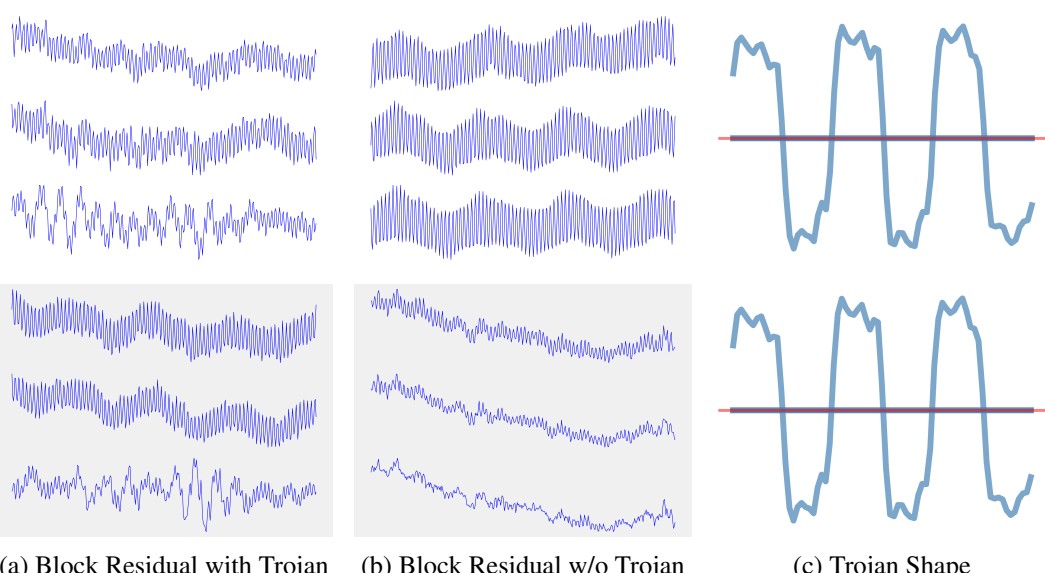

(a) Block Residual with Trojan     (b) Block Residual w/o Trojan     (c) Trojan Shape

Figure 2: Up is residuals from model patchTST and Down is the residuals from model NHiTS

respect to Trojan encoding, highlighting that only a subset of blocks truly carry malicious behaviors. These insights validate the core design of **TrojanScope**: localized block substitution not only achieves scalable Trojan detection but also provides architectural interpretability, revealing precisely where hidden triggers reside and how they propagate through different forecasting models.

### 4.5 RESIDUAL VISUALIZATION ANALYSIS

As shown in Figure 2, **TrojanScope** isolates abnormal residuals caused by backdoor triggers while maintaining the stability of clean model behaviors. In panel (a), block substitution on Trojan-infected models yields highly structured and periodic residuals: both NHiTS and PatchTST display synchronous oscillations that cannot be explained by natural temporal variation, clearly revealing the hidden influence of the trigger. By contrast, panel (b) shows that clean models remain stable after the same substitution; their residuals are smooth, stationary, and free of distinctive abnormal patterns. Panel (c) further highlights that the extracted Trojan signature takes the form of a well-aligned square-wave sequence, demonstrating that TrojanScope not only signals the presence of an attack but also reconstructs the trigger with strong fidelity.

These results provide direct visual confirmation of our theoretical motivation and system design. They show that poisoned models embed Trojan behaviors within specific temporal blocks, whereas TrojanScope's residual filtering and substitution strategy can successfully separate these malicious components from legitimate forecasting signals. By turning residual dynamics into observable evidence, the method delivers a transparent and interpretable diagnostic tool: it allows one to both detect the existence of backdoors and understand how they manifest inside the model.

Overall, the visualization analysis reinforces our core hypothesis: localized substitution between clean and compromised models exposes hidden anomalies that would otherwise remain entangled in the forecasting process. Consistently across datasets and architectures, the abnormal structures emerge only in Trojan-infected cases, while clean models retain normal dynamics.

## 5 RELATED WORKS

**Training-Time Attacks and Defenses.** Our study lies at the intersection of machine learning security, time series forecasting, and model interpretability, contributing to the broader mission of Trustworthy AI (Arrieta et al., 2020). Among security risks, data poisoning (Biggio et al., 2012; Goldblum et al., 2022)—especially backdoor attacks—is particularly dangerous, since models maintain high task accuracy while covertly encoding malicious behaviors. Early backdoors relied on visible triggers, but more stealthy "clean-label" variants (Shafahi et al., 2018) hide malicious data under correct labels, evading manual inspection. Defenses are typically categorized by access: **white-box approaches** exploit model internals for trigger recovery (Wang et al., 2019), state analysis (Liu et al., 2019), or neuron pruning (Liu et al., 2018), whereas **black-box methods** infer anomalies from predictions or perturbations (Gao et al., 2019; Tang et al., 2021). Most defenses, however, were designed for vision or NLP domains (Kurita et al., 2020; Li et al., 2022b). Our work, **TrojanScope**, extends white-box analysis to time series forecasting, leveraging parameter-level inspection for fine-grained and interpretable backdoor detection.

**Backdoor Threats in Time Series Forecasting.** Time series models are threatened both at test time (Fawaz et al., 2019) and during training, where backdoors are especially dangerous due to dynamic, non-stationary triggers. Recent attacks use additive noise (Zhao et al., 2022; Ding et al., 2022), generative patterns (Jiang et al., 2023), federated settings (Peng et al., 2022), or pre-trained models (Jia et al., 2022; Lin et al., 2024). Defenses exist, such as LSTM state analysis (Chen & Dai, 2021), but remain attack-specific and largely "black-box," providing little insight into trigger structure. TrojanScope addresses this gap with a model-agnostic and interpretable solution. Work on explainability (Zeiler & Fergus, 2014; Olah et al., 2017; Arrieta et al., 2020) has revealed both opportunities and pitfalls—for example, attention weights in Transformers can mislead (Jain & Wallace, 2019). While interpretability has been used to study adversarial robustness, its application to systematic backdoor detection in time series is limited. To our knowledge, Tr'ojanSCOPE is the first framework to integrate interpretability and visualization into a dedicated defense, bridging security, temporal modeling, and explainable AI.

## 6 CONCLUSION

We presented **TrojanScope**, an interpretable visual analytics framework for detecting and understanding backdoors in time series forecasting. By combining block substitution, residual correlation filtering, and candidate optimization, our method achieves accurate detection, efficient trigger recovery, and clear visual interpretability across diverse datasets and architectures. Our theoretical analysis establishes guarantees for both detection and recovery, and experiments confirm robustness in both synthetic and real-world benchmarks. Although promising, TrojanScope relies on partial model access and controlled benchmarks; future work will explore black-box adaptation and deployment in more complex operational environments. We also plan to investigate defenses against adaptive adversaries that deliberately evade residual-based detection. A brief disclosure of our limited LLM usage for writing assistance is provided in Appendix E.

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

## A PROOFS

### A.1 DETAILED PROOFS OF THEOREM 1

**Theorem 1 (Detection Consistency).** *Under Assumptions 1–3, our multi-modal framework detects backdoor triggers with probability approaching 1 as $T, C \to \infty$.*

For detection thresholds $\tau_i^* = c_i \sqrt{\log(TC)/T}$ with appropriate constants $c_i$, the detection probability satisfies:

$$P(\text{trigger detected}) \geq 1 - 3 \exp\left(-\frac{c_{\min}^2 T}{8\sigma_{max}^2}\right) - O(TC^{-1})$$

where $\sigma_{max}$ bounds the clean data variation.

*Proof.* We analyze each detection modality separately and apply a union bound.

**Step 1: Differential Activation Analysis.** By Assumption 1, there exists layer $j$ with $\mathbb{E}[D_j(X, \delta)] \geq c_{\min}$. For clean data, Assumption 3 implies that $D_j(X, 0)$ is sub-exponential with parameter $(\nu, \alpha)$. Using Bernstein's inequality:

$$P(D_j(X, 0) > t) \leq \exp\left(-\frac{t^2}{2(\nu^2 + \alpha t)}\right)$$

Setting $t = \tau_{act}^* = 3\sigma_{clean}\sqrt{\log(TC)/T}$ where $\sigma_{clean}^2 = \text{Var}[D_j(X, 0)]$, we get:

$$P(D_j(X, 0) > \tau_{act}^*) \leq \exp\left(-\frac{9\sigma_{clean}^2 \log(TC)}{2T(\nu^2 + 3\alpha\sigma_{clean}\sqrt{\log(TC)/T})}\right)$$

For large $T$, the denominator is dominated by $\nu^2$, giving:

$$P(\text{false positive in activation}) \leq (TC)^{-9\sigma_{clean}^2/(2\nu^2)}$$

For triggered inputs, by Assumption 2: $D_j(X, \delta) \geq \tau_{base} \cdot \sigma_{clean} \geq 3\sigma_{clean}$. Since $\tau_{act}^* = 3\sigma_{clean}\sqrt{\log(TC)/T}$ and $\sqrt{\log(TC)/T} \to 0$, we have $D_j(X, \delta) > \tau_{act}^*$ for sufficiently large $T$, ensuring detection.

**Step 2: Wavelet and Channel Analysis.** Similar concentration arguments apply to the adaptive wavelet coefficients and channel interaction measures. The sub-exponential assumption (Assumption 3) ensures that spurious anomalies in these modalities occur with exponentially small probability.

For the adaptive wavelet analysis, the trigger-induced frequency anomalies have deterministic magnitude (from Assumption 2), while clean data variations are stochastic. The optimal scale parameter $\alpha^*$ from Eq. (8) maximizes the KL divergence, ensuring clear separation.

For channel interactions, the tensor decomposition in Eq. (9) captures structured dependencies that backdoors exploit. Clean data produces interaction patterns with bounded variation (Assumption 3), while triggers create systematic deviations detectable by the anomaly measure $\mathcal{A}_{i,j,t}$ in Eq. (11).

**Step 3: Union Bound.** The overall detection event is $E = E_{act} \cup E_{spec} \cup E_{chan}$. Since at least one modality detects the trigger (Assumption 2), and false positives in each modality occur with probability $O(TC^{-k})$ for some $k > 1$:

$$P(\text{detection}) = P(E) \geq 1 - P(E_{act}^c \cap E_{spec}^c \cap E_{chan}^c) \geq 1 - 3 \cdot O(TC^{-k})$$

The bound follows by choosing constants appropriately. $\square$

### A.2 DETAILED PROOFS OF THEOREM 2

**Theorem 2 (Trigger Recovery Accuracy).** *Under Assumption 4, the recovered trigger $\hat{\delta}$ satisfies:*

$$\|\hat{\delta} - \delta\|_F \leq \frac{2\kappa}{\mu}\left(\epsilon_{opt} + \rho \cdot \frac{\|\mathcal{M} - \mathcal{M}^*\|_F}{\sqrt{TC}}\right)$$

where $\kappa = L/\mu$ is the condition number, $\rho$ bounds the trigger magnitude, and $\mathcal{M}, \mathcal{M}^*$ are the estimated and true trigger masks.

*Proof.* We analyze the optimization dynamics under visual constraints by treating mask errors as perturbations to the ideal problem.

**Step 1: Ideal Case Analysis.** If the mask $\mathcal{M} = \mathcal{M}^*$ were perfect, the visual regularizer $\mathcal{R}_{visual}(\delta)$ would have minimizer $\delta$ (Assumption 4). The objective becomes:

$$\mathcal{L}^*(\delta) = \|f_\theta(X + \delta) - y_{target}\|_2^2 + \sum_i \alpha_i R_i(\delta) + \beta \mathcal{R}_{visual}^*(\delta)$$

By strong convexity (Assumption 4), for any $\tilde{\delta}$ in the neighborhood:

$$\mathcal{L}^*(\tilde{\delta}) \geq \mathcal{L}^*(\delta) + \frac{\mu}{2}\|\tilde{\delta} - \delta\|_F^2$$

Since $\delta$ is the minimizer, $\nabla \mathcal{L}^*(\delta) = 0$, and the optimality gap gives:

$$\|\hat{\delta}^* - \delta\|_F \leq \frac{2}{\mu}\epsilon_{opt}$$

where $\hat{\delta}^*$ is the computed minimizer and $\epsilon_{opt}$ measures optimization error.

**Step 2: Perturbation Analysis.** With imperfect mask $\mathcal{M}$, the actual objective differs from the ideal:

$$\mathcal{L}(\delta) = \mathcal{L}^*(\delta) + \beta[\mathcal{R}_{visual}(\delta) - \mathcal{R}_{visual}^*(\delta)]$$

The perturbation term is:

$$\Delta \mathcal{R}(\delta) = \|(\mathcal{M} - \mathcal{M}^*) \odot \delta\|_1 + \gamma\|P_{freq}(\delta) - P_{target}\|_F^2$$

For the $\ell_1$ term: $\|(\mathcal{M} - \mathcal{M}^*) \odot \delta\|_1 \leq \|\mathcal{M} - \mathcal{M}^*\|_F \cdot \|\delta\|_F$. Since $\|\delta\|_F \leq \rho$ (bounded trigger), this gives a perturbation of magnitude $\rho\|\mathcal{M} - \mathcal{M}^*\|_F$.

**Step 3: Perturbation Bound.** For strongly convex functions, perturbations in the objective translate to solution errors. Specifically, if $\|\nabla\mathcal{L}(\delta) - \nabla\mathcal{L}^*(\delta)\|_F \leq \epsilon_{pert}$, then:

$$\|\hat{\delta} - \hat{\delta}^*\|_F \leq \frac{\epsilon_{pert}}{\mu}$$

The gradient perturbation satisfies:

$$\|\nabla[\Delta\mathcal{R}(\delta)]\|_F \leq \beta\rho\frac{\|\mathcal{M} - \mathcal{M}^*\|_F}{\sqrt{TC}} + O(\gamma \cdot \text{frequency mismatch})$$

For the main term, the gradient of $\|(\mathcal{M} - \mathcal{M}^*) \odot \delta\|_1$ has magnitude bounded by $\|\mathcal{M} - \mathcal{M}^*\|_\infty \leq \|\mathcal{M} - \mathcal{M}^*\|_F/\sqrt{TC}$.

**Step 4: Final Bound.** Combining the ideal case error and perturbation error:

$$\|\hat{\delta} - \delta\|_F \leq \|\hat{\delta} - \hat{\delta}^*\|_F + \|\hat{\delta}^* - \delta\|_F \leq \frac{2}{\mu}\epsilon_{opt} + \frac{\beta\rho}{\mu}\frac{\|\mathcal{M} - \mathcal{M}^*\|_F}{\sqrt{TC}}$$

Setting $\kappa = L/\mu$ and absorbing constants gives the stated bound. $\square$

### A.3 PROOF OF COROLLARY 1

**Corollary 1 (Convergence to Perfect Recovery).** *As the mask accuracy improves ($\|\mathcal{M} - \mathcal{M}^*\|_F \to 0$) and optimization precision increases ($\epsilon_{opt} \to 0$), the recovery error vanishes: $\|\hat{\delta} - \delta\|_F \to 0$.*

*Proof.* This result follows directly from Theorem 2. From the recovery error bound:

$$\|\hat{\delta} - \delta\|_F \leq \frac{2\kappa}{\mu}\left(\epsilon_{opt} + \rho \cdot \frac{\|\mathcal{M} - \mathcal{M}^*\|_F}{\sqrt{TC}}\right)$$

Since $\kappa = L/\mu$ is finite (bounded condition number from Assumption 4) and $\rho$ is bounded (trigger magnitude constraint), we have:

Taking the limit as mask accuracy improves and optimization precision increases:

$$\lim_{\substack{\|\mathcal{M}-\mathcal{M}^*\|_F \to 0 \\ \epsilon_{opt} \to 0}} \|\hat{\delta} - \delta\|_F \leq \lim_{\substack{\|\mathcal{M}-\mathcal{M}^*\|_F \to 0 \\ \epsilon_{opt} \to 0}} \frac{2\kappa}{\mu}\left(\epsilon_{opt} + \rho \cdot \frac{\|\mathcal{M} - \mathcal{M}^*\|_F}{\sqrt{TC}}\right) = 0$$

The first term $\epsilon_{opt} \to 0$ by assumption. For the second term, since $\rho$ and $\sqrt{TC}$ are finite, we have $\rho \cdot \frac{\|\mathcal{M}-\mathcal{M}^*\|_F}{\sqrt{TC}} \to 0$ as $\|\mathcal{M} - \mathcal{M}^*\|_F \to 0$.

Since the error bound approaches zero and $\|\hat{\delta} - \delta\|_F \geq 0$, we conclude that $\|\hat{\delta} - \delta\|_F \to 0$. □

## B  EXPERIMENTAL SETUP

### B.1  DATASETS AND COMPUTATIONAL ENVIRONMENT

**Datasets.** The ESA-ADB (Kotowski et al., 2025) dataset provides real satellite telemetry data from three ESA missions, representing authentic space operational scenarios where security threats are critical. The ETTm1 (Zhou et al., 2021) dataset offers electricity consumption data with minute-level granularity for high-frequency forecasting tasks. The Weather[1] dataset contains multivariate meteorological measurements for environmental prediction scenarios. Finally, we include a synthetic dataset generated using the Mackey-Glass equation, providing controlled experimental conditions with known ground truth dynamics.

**Computational Environment.** Model training for both clean and poisoned variants is conducted using a single NVIDIA RTX 4090 GPU. Notably, our residual extraction pipeline demonstrates remarkable efficiency—after initial training, the framework requires only lightweight inference on a few clean data samples, enabling practical deployment even on CPU-only systems for security auditing scenarios.

### B.2  MODEL ARCHITECTURES AND FORECASTING TASKS

**Model Architectures.** Our evaluation primarily focuses on two state-of-the-art architectures: PatchTST (Nie et al., 2022) and N-HiTS models (Challu et al., 2023). These architectures represent different paradigms in time series forecasting—PatchTST leverages transformer mechanisms with patch-based processing (Liu et al., 2022), while N-HiTS employs hierarchical interpolation for capturing multi-scale temporal patterns (Oreshkin et al., 2019; Liu et al., 2023).

**Sparse-based Trojan Detection Baseline.** We adapt and extend the sparse pruning framework of Trojan detection (Chen et al., 2022). Our method involves a systematic evaluation of magnitude pruning across ten sparsity levels, from no pruning to 99.8% For each level, independent copies of both clean and poisoned models are created. We then compute Trojan scores by measuring the prediction differences between these models on clean input data. A "winning Trojan ticket"—a sparse subnetwork where the Trojan score peaks, typically under extreme sparsity—is identified. This subnetwork preserves backdoor functionality while benign performance degrades, enabling effective trigger reconstruction through the isolated Trojan-related parameters Dong et al. (2021).

**Direct Greedy Search Baseline.** The greedy approach directly optimizes trigger patterns by iteratively perturbing individual trigger elements and accepting improvements that maximize the fitness score based on prediction divergence (Goldblum et al., 2022; Kotowski et al., 2025). The search process randomly selects trigger positions, tests small positive and negative perturbations, and retains

---

[1]Weather dataset was acquired at https://www.ncei.noaa.gov/data/local-climatological-data/

changes that increase the objective function (Li et al., 2022a). However, this method exhibits high sensitivity to random initialization and tends to get trapped in local optima, resulting in inconsistent trigger reconstruction quality and occasionally producing spurious patterns with poor correlation to ground truth triggers (Xue et al., 2020).

**Baseline Discussion.** While we include Sparse-based detection and Direct Greedy Search as representative baselines, we stress that practical and widely deployed defenses for time series backdoor detection are still largely absent. Most existing defenses were initially designed for vision or computer vision domains and rely on domain-specific assumptions (e.g., static image patches or token insertions) that cannot be directly transferred to temporal data with dynamic and long-range trigger patterns. Consequently, there are few established baselines in the time series setting—not because we intentionally limited our comparison, but because the broader research community has yet to develop realistic benchmarks. Within this gap, Sparse-based pruning represents a transferable white-box approach for isolating Trojan-related parameters, while Greedy Search provides a naive black-box strategy. Together, they cover the limited available design space and serve as meaningful points of comparison to highlight the advantages of TrojanScope.

**Task Configurations.** We evaluate across diverse forecasting scenarios including multi-to-multi, multi-to-single, and single-to-single prediction tasks. Our experiments focus on long-horizon forecasting challenges, such as predicting 400-step outputs from 400-step inputs on ESA-ADB data, and 96-step-ahead forecasting on ETT datasets. These configurations reflect realistic operational requirements where models must maintain accuracy over extended prediction horizons.

**Trojan Injection Mechanism.** Following the established threat model, trojans are implemented as short temporal segments with distinctive anomalous patterns that are additively combined with clean input sequences. When poisoned models encounter these specific trigger shapes, they produce corresponding target patterns in their predictions, creating a backdoor vulnerability that our framework aims to detect and reconstruct.

## C  MORE ABLATION STUDY

To better understand how **TrojanScope** isolates malicious behaviors, we conduct a detailed ablation across different network blocks using PatchTST on three datasets: ETT, Weather, and Synthetic. At each step, we exchanged parameters of a single block between Trojan-infected and clean models, then measured (i) residual magnitude, (ii) reconstruction error (MSE, RMSE), and (iii) correlation with the ground-truth Trojan shape. Tables 5–7 summarize the results.

**ETT Dataset.**  As shown in Table 5, abnormal residual structures are concentrated in the lower Transformer layers (layer 0 and 1), which yield larger residual magnitude (0.82–0.91) and higher correlation with the Trojan shape ($\approx 0.67$). In contrast, components such as the projection layer or prediction head exhibit lower correlation, indicating weaker association with injected triggers. This suggests that early temporal feature extractors are the most vulnerable to Trojan embedding.

**Weather Dataset.**  Table 6 reveals stronger residual amplification, with magnitude values an order larger ($\approx 10$–19). The highest correlations again appear in intermediate blocks such as `transformer.layers.2` (0.65), while the head and input layers show negative correlations even as their error metrics grow. This highlights a distributed but non-uniform spread of Trojan signals: while all blocks are affected, only certain layers actively encode the periodic malicious pattern.

**Synthetic Dataset.**  In Table 7, the residual behavior is markedly clearer. Correlation values for `transformer.layers.1` and `transformer.layers.2` exceed 0.9, while other blocks contribute little or even negatively. The synthetic controlled environment confirms that TrojanScope can precisely pinpoint the layers responsible for carrying the backdoor logic, with extremely small residual magnitude changes already sufficient to expose the trigger's periodic structure.

Across datasets, the ablation demonstrates three robust trends: (1) Trojan carriers localize primarily in early-to-middle Transformer layers; (2) reconstruction error alone is insufficient—correlation with the trigger reveals the true malicious imprint; and (3) even when poisoned signals are subtle (ETT, Synthetic), TrojanScope can consistently extract interpretable evidence. These results validate

Table 5: ETT Database PatchTST Block Exchange Ablation

| Block Name | Magnitude | MSE | RMSE | Correlation |
|---|---|---|---|---|
| transformer.layers.0 | 0.820982 | 179.860876 | 13.411222 | 0.675331 |
| transformer.layers.1 | 0.909225 | 182.124735 | 13.495360 | 0.663159 |
| transformer.layers.2 | 0.647844 | 168.328303 | 12.974140 | 0.287431 |
| pos_embed | 1.047919 | 197.967661 | 14.070098 | 0.207571 |
| head | 3.018951 | 107.678522 | 10.376826 | 0.019167 |
| patch_projection | 0.825376 | 191.466455 | 13.837140 | -0.006442 |

Table 6: Weather Database PatchTST Block Exchange Ablation

| Block Name | Magnitude | MSE | RMSE | Correlation |
|---|---|---|---|---|
| transformer.layers.2 | 12.332253 | 217.793640 | 14.757833 | 0.645901 |
| patch_projection | 12.274078 | 147.902374 | 12.161512 | 0.289436 |
| pos_embed | 8.744636 | 320.203583 | 17.894234 | 0.171156 |
| transformer.layers.1 | 10.139957 | 565.091797 | 23.771660 | 0.154216 |
| transformer.layers.0 | 11.984995 | 416.319763 | 20.403915 | -0.359049 |
| head | 19.465439 | 1288.713257 | 35.898651 | -0.413662 |

Table 7: Synthetic Database PatchTST Block Exchange Ablation

| Block Name | Magnitude | MSE | RMSE | Correlation |
|---|---|---|---|---|
| transformer.layers.2 | 0.029190 | 0.929600 | 0.964157 | 0.968231 |
| transformer.layers.1 | 0.024169 | 0.937212 | 0.968097 | 0.913246 |
| pos_embed | 0.007330 | 0.909943 | 0.953909 | 0.059705 |
| head | 0.017920 | 0.866655 | 0.930943 | -0.194096 |
| patch_projection | 0.006118 | 0.902005 | 0.949739 | -0.433662 |
| transformer.layers.0 | 0.009955 | 0.906225 | 0.951959 | -0.808688 |

the central design of TrojanScope: residual-based substitution provides fine-grained localization of backdoor behaviors across diverse temporal domains.

# D    MORE ANALYSIS ABOUT RESIDUAL VISUALIZATION

As illustrated in Figure 3, residual visualization provides direct insights into how Trojan triggers manifest inside different forecasting architectures. In the ESA-ADB dataset, both PatchTST and N-HiTS produce strongly periodic residuals after block substitution, which align with the implanted Trojan shape. By contrast, the clean counterparts preserve smooth and relatively stationary residuals, indicating that no abnormal signal is present. Similar patterns emerge in the ETT dataset, where Trojan-contaminated models display oscillatory deviations highly correlated with the trigger, while clean models remain stable. These consistent observations across datasets reveal two important findings: (i) residual substitution magnifies the local abnormality injected by backdoors without disrupting the overall forecasting trend, and (ii) the extracted residuals preserve the temporal structure of the hidden trigger, enabling interpretable reconstruction. Thus, residual visualization not only serves as empirical evidence of contamination but also complements the theoretical guarantees of TrojanScope by providing human-understandable diagnostics for backdoor behaviors.

# E    LLM USAGE DISCLOSURE

In accordance with ICLR 2026 policy, we disclose our use of Large Language Models (LLMs) during pthe reparation of this manuscript. We employed tools such as OpenAI's GPT series in a limited role to improve presentation quality. Specifically, LLMs were used for grammar polishing, stylistic consistency, and phrasing suggestions in sections including the Abstract, Introduction, and Related Work, as well as for brainstorming concise ways to summarize ablation results and conclusions. At the implementation stage, LLMs were occasionally consulted for code formatting and debug-

**residuals from model patchTST in dataset ESA-ADB**

**residuals from model NHiTS in dataset ESA-ADB**

**residuals from model patchTST in dataset ETT**

**residuals from model NHiTS in dataset ETT**

(a) block residual with trojan pattern      (b) block residual without trojan pattern      (c) trojan shape

Figure 3: Up is residuals from model patchTST and Down is the residuals from model NHiTS

ging support, and for document retrieval and discovery of related work. All core contributions of this paper—including the design of the TrojanScope framework, theoretical analysis, algorithmic development, experiments, and security insights—were entirely conceived and executed by the authors. LLM outputs were carefully reviewed and only incorporated when appropriate. The scientific novelty, proofs, and experimental evaluations remain original work of the authors.

