# OpenReview forum: "TrojanScope: Interpretable Backdoor Detection for Time Series Forecasting"
_ICLR.cc/2026/Conference — ICLR 2026 Conference Withdrawn Submission_

### Official Review · Reviewer_9TzZ · 2025-10-27

**Soundness:** 2
**Presentation:** 3
**Contribution:** 2
**Rating:** 4
**Confidence:** 4

**Summary:**

This paper proposes TrojanScope, a novel visual analytics framework for detecting and interpreting backdoor attacks in time series forecasting models. TrojanScope's integrates three main stages: (i) a block-level substitution strategy to localize contaminated components within the model by creating hybrid clean-poisoned models, (ii) residual filtering based on correlation analysis to isolate abnormal temporal channels affected by the trigger, and (iii) a candidate extraction and optimization process to reconstruct the trigger's pattern. The authors also provide theoretical guarantees for detection consistency and recovery accuracy.

**Strengths:**

- The paper is well-written and easy to follow.
- The paper raises a timely and significant problem: backdoor detection specifically tailored for time series forecasting, a domain where existing defenses are lacking.

**Weaknesses:**

- The most significant weakness is the core assumption that the defender has access to a corresponding clean model. The entire methodology heavily relies on a trusted clean counterpart. This is a very strong assumption that is unlikely to hold in many practical scenarios (e.g., downloading a single pre-trained model from a public hub). The paper does not sufficiently discuss the feasibility of this or propose alternatives if a clean model is unavailable.
- The evaluation appears limited to additive triggers (as per Section 4.1). It is unclear how TrojanScope would perform against other, potentially more stealthy, backdoor attack types, such as warping-based triggers, clean-label attacks, or triggers that are non-linearly integrated into the input data. The method's reliance on residual analysis might be less effective if the trigger doesn't create a distinct additive or separable artifact.
- The "block-level substitution" approach, while effective on the tested models (PatchTST, N-HiTS), seems dependent on a modular stack-based architecture. It is not clear how this would generalize to other common time series models (e.g., traditional RNNs/LSTMs, linear models) that lack such a clearly partitionable "block" structure.
- The methodology introduces several new hyperparameters (e.g., correlation threshold $\tau$ in Eq. 5, scoring weights $\alpha, \beta, \gamma$ in Eq. 8, window/stride parameters for candidate extraction) without a clear ablation study or sensitivity analysis. It is unknown how robust the method is to different choices of these parameters or how a practitioner would reliably set them for a new, unseen model or attack.
- The benchmarks used in this paper are limited. For example, the Weather dataset contains numerous default values that remain unprocessed, making it nearly impossible for most methods to generate accurate predictions. The authors should consider more reliable benchmarks to strengthen their work, such as the full evaluation on ETT datasets (i.e., ETTh1, ETTh2, ETTm1, ETTm2), Traffic dataset (e.g., PeMS03, PeMS04, PeMS07, PeMS08). The author can also refer to the multiple benchmarks and baselines recommended in this position work [7], such as KDDCup2018. There is also lack of recent time series forecasting baselines [1-6]. Authors should consider more different architectures to enhance their robustness.


Reference:
[1] Liu, Minhao, et al. "Scinet: Time series modeling and forecasting with sample convolution and interaction." arXiv preprint arXiv:2106.09305 (2021).

[2] Das, Abhimanyu, et al. "Long-term forecasting with tide: Time-series dense encoder." arXiv preprint arXiv:2304.08424 (2023).

[3] Wang, Shiyu, et al. "TimeMixer: Decomposable Multiscale Mixing for Time Series Forecasting." arXiv preprint arXiv:2405.14616 (2024).

[4] Zhang, Yunhao, et al. "Crossformer: Transformer Utilizing Cross-Dimension Dependency for Multivariate Time Series Forecasting." International Conference on Learning Representations (2023).

[5] Li, Zhe, et al. "Revisiting Long-term Time Series Forecasting: An Investigation on Linear Mapping." arXiv preprint arXiv:2305.10721 (2023).

[6] Wu, Haixu, et al. "TimesNet: Temporal 2D-Variation Modeling for General Time Series Analysis." arXiv preprint arXiv:2210.02186 (2022).

[7] Brigato, Lorenzo, et al. "Position: There are no Champions in Long-Term Time Series Forecasting." arXiv preprint arXiv:2502.14045 (2025).

**Questions:**

My primary question concerns the framework's core assumption of requiring a clean model for comparison. This seems to limit its practical applicability, as defenders often only possess the single suspect model. Could the authors elaborate on scenarios where this assumption holds? Furthermore, have the authors considered or explored methods to approximate a 'clean' reference model? For example, could a 'clean' version be generated from the poisoned model itself, perhaps through techniques like fine-tuning on a small trusted dataset, model pruning, or parameter editing?

---

### Official Review · Reviewer_BhJ5 · 2025-10-31

**Soundness:** 3
**Presentation:** 2
**Contribution:** 2
**Rating:** 4
**Confidence:** 3

**Summary:**

The paper proposes a practical pipeline to detect and localize backdoor behavior in time-series forecasters by swapping blocks between a known-clean and a suspect model and watching for spikes in residuals. It then narrows suspects via temporal, spectral, and channel checks and attempts trigger recovery with masked optimization. The authors provide theoretical guarantees under strong assumptions and demonstrate results on PatchTST and N-HiTS.

**Strengths:**

- Block-swap + residual-spike procedure is easy to implement.
- Interpretability: Residual line plots, per-block spike scores (e.g., \(V_b\), \(D_b\)), and block-swap tables that surface suspicious blocks.
- Uses temporal, spectral, and channel consistency to reduce false alarms.
- Covers two architectures: Works on a transformer (PatchTST) and a non-transformer (N-HiTS).
- Includes a trigger recovery step and offers theory with stated assumptions.

**Weaknesses:**

- The method assumes you have both a known-clean model and the suspicious model so you can swap blocks between them. In real life, you often only have the suspicious model. Without a black-box version, this limits where the method can be used.

- The approach looks for a sharp change when swapping one block. If the malicious behavior is spread across several blocks each with a small effect, the “spike” may never appear, so the method could miss it.

- The PatchTST score compares the current residual to a “previous” residual, which is the one from the immediately prior block swap, which makes the score dependent on the swap order and normalization. The N-HiTS variant similarly relies on hand-chosen comparisons (single vs. cumulative residuals). The paper does not report sensitivity analyses for these choices, so it’s unclear how much conclusions change with a different ordering or scaling.

- Detection assumes the Trojan signal clearly stands out from normal noise, and that the clean residuals behave nicely such as light tails, weak dependence. The recovery guarantee assumes the objective is locally strongly convex and smooth around the true trigger. These are strong assumptions for deep time-series models and may not hold in practice.

- The pipeline uses a correlation threshold and a set of weighting coefficients to score candidates, but there’s no ablation showing how results change if you tweak them. That makes the method feel brittle and hard to reproduce.

**Questions:**

See weakness

---

### Official Review · Reviewer_t35w · 2025-10-31

**Soundness:** 2
**Presentation:** 2
**Contribution:** 2
**Rating:** 2
**Confidence:** 4

**Summary:**

This paper introduces TrojanScope, a framework for detecting and understanding backdoor attacks in time series forecasting models. It combines block substitution, residual filtering, and candidate optimization to localize and reconstruct hidden triggers, with theoretical guarantees for detection and recovery. Experiments on a few datasets (ESA-ADB, ETTm1, Weather, Synthetic) and two models (N-HiTS, PatchTST) show seemingly good performance.

**Strengths:**

The idea of having an interpretable time series backdoor detection is interesting. Some detection consistency and convergence analyses were provided.

**Weaknesses:**

1. The introduction is generally well-written and did its job in introducing the problem setting. But the content after that starts to get harder to follow. Many design choices are just being thrown out with little justification or discussion, which really confuse me a lot. For example, when I read line 134-143, I wonder why only PatchTST and N-HiTS are the two models used? Is it because they are the SOTA (as far as I know they are probably not anymore), or they are just popular, or the TrojanScope is specifically designed for them due to some specific reasons? No matter which is the case, I would suggest the author to either add such discussions, preferably before suddenly introducing them in Methodology, to justify your design choices.

2. Also, new notations/symbols are being introduced whenever it is needed in the context. I’m not saying this is always bad but it generally makes it hard for readers to catch the key point or intuition of each equations (because every one of them contains new notations). For this I suggest the author have a preliminary (sub)section for formalizing the problem and defining the organized & unified notations.

3. Following 1, the authors also only used NHiTS and PatchTST in their experiments, and only tested on 4 (1 of them is synthetic) time series datasets. I understand that training both healthy and poisoned models for each dataset could be costly, but the generality to different dataset and models is just important for any methods, and I would expect to see at least one (even with less comprehensive setting) experiment to validate the versatility of the proposed method on more datasets and models.

4. Some additional suggestions: polish the table and figure titles, make them self-contained if possible, e.g., the figure 2 title is simply describing the layout rather than the figure itself. It’s hard to understand for readers who are skimming and scanning the paper. I think a good paper can be made out of this topic but there is a large improvement room left for this manuscript.

**Questions:**

Please see the weaknesses, where I also gave my suggestions.

---

### Official Review · Reviewer_FQuy · 2025-11-01

**Soundness:** 2
**Presentation:** 4
**Contribution:** 2
**Rating:** 2
**Confidence:** 4

**Summary:**

This submission focuses on backdoor detection in time series forecasting. It highlights three research challenges: the natural temporal complexity in time series, the complicated new forecasting mechanism, and the lack of interpretation. To address these issues, the submission proposes TrojanScope,  which leverages multi-scale temporal analysis and channel interaction graphs to reveal how backdoor triggers propagate through neural networks.

**Strengths:**

**Interesting Research Topic:** The study addresses an important and timely problem, identifying backdoors in time series forecasting models. This topic is both novel and significant, as the security and robustness of temporal models have received far less attention compared to image and text domains.

**Comprehensive Contribution:** The submission not only presents the design and experimental validation of the proposed approach but also includes a theoretical analysis, which enhances the depth and credibility of the work. The integration of both empirical and analytical perspectives strengthens the overall contribution.

**Weaknesses:**

- **Major assumption flaw:** The framework appears to assume access to **clean/poisoned model labels**. This is fundamentally unrealistic in practice. Users typically do not know whether a forecasting model has been backdoored, which is precisely why backdoor detection is needed.

- **Limited generalization:** The method seems tailored to **N-HiTS** and **PatchTST**, with no clear evidence that it extends to broader time-series architectures. This limits the contribution and impact.

- **Marginal gains over baselines:** Compared with baselines (e.g., **Sparse**), the reported improvements are modest, making the net technical advance unclear.

- **Methodological/experimental opacity:** Many implementation details and experimental settings are insufficiently specified (see Questions section), hindering reproducibility and interpretability.

**Questions:**

i. **Labeling assumption:** Are models in your setup labeled as “clean” or “poisoned”? If so, this assumption is unrealistic; practitioners typically do not know a model’s status. Please clarify.

ii. **Unlabeled models:** If models are unlabeled, how does TrojanScope avoid pairing two clean models or two poisoned models during its comparison step?

iii. **Method scope (Sec. 2.2):** Is the residual-filtering approach in Section 2.2 specific to N-HiTS? How would you implement residual filtering for PatchTST?

iv. **Generalization:** Does TrojanScope generalize to other forecasting architectures beyond PatchTST and N-HiTS? If not, please state limitations.

v. **TrojanShape illustration:** Provide a clearer figure for TrojanShape with a legend and brief explanation of each curve/component.

vi. **Trigger recovery:** Equation (8) identifies suspicious patterns, but what is the concrete procedure to recover the actual trigger from the detected score?

vii. **Multiple triggers:** Can a single poisoned model contain multiple distinct triggers? If so, can your method detect and recover multiple triggers?

viii. **Cross-dataset poisoned models:** The poisoned model used in experiments is trained on ESA-ADB. How would you obtain or simulate poisoned models for other datasets (e.g., weather) in a realistic way?

ix. **Visualization:** Figure 2 appears unclear. What dataset is being used? Which network block is the residual computed from? Additionally, please show how the block-wise residuals look to clarify where and how the differences are measured.

x. **Interpretation:** Why are the attacks encoded within a single block instead of resulting from the accumulated effect of multiple blocks? Please provide an explanation or empirical evidence supporting this design choice. Are the attacks concentrated in a single block due to an inherent natural feature of the model/bookdoor attacks, or is this pattern caused by the critical block identification process introduced in the method? Please clarify the underlying reason.

**References**
[1] Kotowski, Krzysztof, et al. "Trojan horse hunt in time series forecasting for space operations." arXiv preprint arXiv:2506.01849 (2025).

---

### Note · Authors · 2026-01-14

**Comment:**

Following thorough discussions with all co-authors, we have decided to withdraw our submitted paper.

**Withdrawal Confirmation:**

I have read and agree with the venue's withdrawal policy on behalf of myself and my co-authors.